# Self-management of chronic conditions including multimorbidity in sub-Saharan Africa: A systematic and meta-synthesis review with focus on diabetes, hypertension, chronic kidney disease, and HIV

**Sangwani Nkhana Salimu**[1,2]*, **Melissa Taylor**[1◉], **Stephen A. Spencer**[2◉],
**Nicola Desmond**[3], **Deborah Nyirenda**[2,3], **Ben Morton**[1,2]

**1** Department of Clinical Sciences, Liverpool School of Tropical Medicine, Liverpool, United Kingdom,
**2** Malawi-Liverpool-Wellcome Programme, Blantyre, Malawi, **3** Department of International Public Health, Liverpool School of Tropical Medicine, Liverpool, United Kingdom

◉ These authors contributed equally to this work.
* sangwani.salimu@lstmed.ac.uk

## Abstract

The increasing prevalence of multimorbidity in sub-Saharan Africa (SSA) is an urgent concern for health service delivery, yet little is known about how best to support self-management- the tasks patients and carers take to maintain physical and mental health in this context. This review synthesized qualitative evidence that describes self-management of four chronic conditions- HIV, diabetes, chronic kidney disease, and hypertension, including multimorbidity among patients and their carers in SSA. We systematically searched five databases and grey literature for studies published between January 2000 and to March 2025 and conducted a thematic synthesis of findings. Twenty-three studies met inclusion criteria, three of which focused on multi-morbidity. Across conditions, patients negotiated self-management based on immediacy of needs and available family support. Patients are motivated to apply biomedical management but are limited by factors such as drug stock-outs and out-of-pocket expenditure. Limited knowledge and low self-efficacy toward self-management of multimorbidity impact decision making and problem solving. We found that diabetes and chronic kidney disease imposed the greatest treatment burden, making them the most challenging conditions for patients to manage. Temporal discontinuation of medications was more prevalent amongst patients with hypertension; and patients with multimorbidity are frequently hypervigilant about their health, more likely to suffer from stress and to seek healthcare. This review synthesised qualitative evidence on self-management of HIV, diabetes, hypertension, and chronic kidney disease in SSA, and considered insights for multimorbidity. Most studies focused on individual conditions, yet our findings reveal strikingly similar challenges across all four conditions: limited health literacy, low self-efficacy, and inadequate structural support. These

**Data availability statement:** All relevant data are within the paper and its Supporting Information files.

**Funding:** This research was funded research was funded by the NIHR (NIHR201708) using UK aid from the UK Government to support global health research. The views expressed are those of the author(s) and not those of the NIHR or the UK government. The views expressed do not reflect the UK government's official policies. The funders had no role in study design, data collection and analysis, decision to publish, or preparation of the manuscript. SAS was supported by a Wellcome Trust Clinical PhD Fellowship (Grant number 203919/Z/16/Z). DN is funded by the Global Health Bioethics Network, a Wellcome Strategic Award (228141/Z/23/Z).

**Competing interests:** The authors have declared that no competing interests exist."

barriers are likely amplified with multimorbidity, further complicating decision-making and self-management. Addressing these gaps will require context-sensitive interventions that strengthen patient literacy, build confidence to increase patient autonomy and expand the range of resources available to manage chronic disease.

## Introduction

Multimorbidity, the presence of two or more long-term conditions (LTCs), is an immediate and increasing health challenge in low- and middle-income countries (LMICs) [1–3]. There are concerns, however, that this definition does not address the person-centered needs of multimorbidity related to the complex relationships between patient or carer preferences, priorities, and social context [2,4,5]. Despite increasing prevalence of multimorbidity, data on patient and carer health literacy, skills and engagement are limited in sub-Saharan Africa (SSA) [3,6]. Self-management refers to the actions individuals and carers take for themselves, their families and others to maintain good physical and mental health; meet social and psychological needs; and prevent illness deterioration [7–9]. This article aims to evaluate available data on the implementation of self-management strategies for multimorbidity in SSA.

Effective self-management requires partnership with healthcare providers to support patients and carers to increase health literacy, skills and confidence [7,8,10,11]. There are notable differences in how multimorbidity presents in SSA compared to high-income countries (HIC); people in SSA more frequently suffer from co-existing chronic non-communicable diseases (NCD) and communicable diseases such as HIV-infection [3,12,13]. This fact, combined with limited access to care and poorly developed pathways for the prevention and management of chronic conditions [14], pose particular challenges for patients. People living with multimorbidity (PLWMM) must constantly negotiate decisions related to their illnesses and interactions with healthcare [15–17]. Individuals self-reflect and self-negotiate on pivotal occurrences related to their conditions; when/where to seek care; and what disease and/or symptoms to prioritize [18,19]. These occurrences function as critical cues or indicators that may lead to significant changes in an individual's approach to self-care, serving as signals that prompt a reassessment and potential refinement of their management strategies. The inter-relationship and recognition of chronic and acute symptoms is a particular challenge in this respect [18,19]. Limited access to healthcare and high out-of-pocket expenditure exacerbates the challenges for PLWMM in SSA [16,20–22]. Furthermore, limited or poor-quality interactions with healthcare providers may also present missed opportunities to improve health literacy and self-management [23].

Our aim was to explore existing evidence on the implementation of self-management strategies for PLWMM, focusing on four common disease combinations (HIV, hypertension, diabetes and chronic kidney disease [CKD]) in SSA [6,24]. A recent systematic review in SSA on the prevalence of individual chronic conditions and multimorbidity among adults admitted to hospital demonstrates

that HIV (36.4%), hypertension (24.4%), diabetes (11.9%), and CKD (7.7%) are the most prevalent NCDs [6]. Their inclusion in self-management interventions in SSA aligns with their high burden and the need for continuous monitoring, medication adherence, and lifestyle modifications [25]. These conditions require active patient engagement in self-care, including blood pressure and glucose monitoring, dietary adjustments, and long-term treatment adherence, making them central to the control of NCDs in SSA. While our review was designed to capture qualitative evidence on multimorbidity, the available literature was limited, with most studies focusing on single conditions. We therefore included studies on each of these conditions individually, while drawing out insights relevant to multimorbidity wherever possible. Lessons drawn from the review will help inform the development of improved patient-centred self-management strategies for PLWMM.

## Methods

### Design

The objective of the qualitative systematic and meta-synthesis review was to explore patient and carer experiences of living with multimorbidity and examine current and potential self-management approaches in SSA. The systematic review protocol was registered in the International Prospective Register of Systematic Reviews (PROSPERO) database (ID: CRD42021262708). We followed methodological and reporting procedures for qualitative meta-synthesis as outlined by Sandelowsi and Barroso (2007) [26].

### Search methods

We conducted a systematic literature search within the following databases: Web of Science, PubMed, CINAHL Complete, MEDLINE Complete, Global Health, Global Health Archive [26]. Our detailed search terms and strategy are described in S1 Table.

### Selection of studies

We included: (1) studies that employed a qualitative, or mixed methodology; (2) explored self-management of multimorbidity among adults (≥18 years old) and/or their carers living with HIV, CKD, diabetes or hypertension or any combination of these diseases; (3) published between January 2000 and March 2025; and (4) conducted in SSA. We excluded (1) abstracts and conference proceedings that were not peer reviewed; (2) studies published in a language other than English; (3) systematic or literature reviews; and (4) studies focused on palliative procedures.

A minimum of two people (SNS, SAS or MT) independently screened titles, abstracts, and full text in turn for manuscript inclusion. Subsequently, any discrepancies were discussed until consensus was reached [26]. We also screened reference lists of papers included in the article. Eligible papers are summarized in S2 Table.

### Quality appraisal

We appraised the quality of eligible papers using the 10-item Critical Appraisal Skills Programme (CASP) checklist for systematic appraisal of qualitative research [27]. SNS and SAS appraised papers independently; discussed discrepancies until they reached consensus or referred to MT for arbitration. We assessed 10 areas- aim, methodology, research design, ethical considerations, research self-scrutiny, recruitment strategy, data collection, data analysis, findings, and study relevance- using a 3-point scale. Studies were rated as Good (7–10), Average (4–6), or Can't Tell (1–3), based on how well they satisfied the criteria. A Good rating indicated the criterion was clearly satisfied; Average indicated some criteria were unmet; and Can't Tell was used when criteria were unclear or not addressed. This approach enabled us to assess study rigor without excluding any. Higher-quality studies (Good) were given greater weight in the synthesis, while those rated Average or Can't Tell were critically examined for methodological limitations but still contributed to the analysis. All

23 included studies described clear aims, utilized appropriate methods, and offered meaningful interpretations, demonstrating strong methodological rigor [26]. Three studies scored Average on three appraisal indicators; however, their overall scores remained high (S3 Table).

## Analysis

Qualitative data synthesis was conducted using a thematic synthesis of qualitative research approach informed by Thomas and Harden, 2008 [28]. This approach is valued for its transparency and systematic structure, which enables researchers to move from descriptive accounts in primary studies to higher-level analytical themes that extend beyond the original findings [28,29]. The first author (SNS) familiarized herself with all the papers through reading and re-reading. SNS then conducted line-by-line coding of the qualitative data, extracting both participant quotations (first-order constructs) and author interpretations as second-order constructs. During this process we excluded quotes from participants who did not have at least one of our target conditions. SNS then generated third-order constructs as themes that represented synthesized interpretations of the data. To increase validity of the synthesis, first order constructs were conceptualized as key concepts, and we then identified quotes that best represented each second-order construct. Additional abstraction led to the development of third-order constructs (new themes), which were discussed amongst the full team (ND, DN and BM). In line with established practice, where necessary, we overrode primary researchers' interpretations in favor of direct participants accounts [30]. We explored codes and examined their relationship to each other to identify themes. Data synthesis was iterative to enhance depth of findings through adding, removing or merging codes as well as re-analyzing the data upon the emergence of new themes.

## Results

### Characteristics of included studies

We screened 2787 papers in the title phase (S4 Table); of these, 95 underwent full screening (S5 Table) and 23 studies fulfilled criteria for inclusion. We reported the process using the PRISMA 2020 checklist to ensure comprehensive reporting of our methodology, including study selection, data extraction, and risk of bias assessment (S1 Fig PRISMA flow chart). Studies reported findings from eight countries: six in South Africa [23,31–35]; five in Malawi [15,36–39]; five in Ghana [40–44] and Liberia [45–47]; two from Nigeria [45,48] and one each from Senegal [49], Ethiopia [50] and Kenya [51]. Five papers purposively recruited PLWMM [15,23,36,43,52] while thirteen studies addressed diabetes [31,33,34,37,39–41,44,46–49,51]. Additionally, three studies covered hypertension [32,45,50] and another two covered HIV [35,36]. The methods used included semi-structured in-depth interviews (IDIs) (n = 12) [15,31,35,36,39,40,42,44,45,48–50]; IDIs and observations (n = 3) [23,33,51]; FGDs (n = 2) [37,43]; IDIs and FGDs (n = 3) [32,34,52]; IDIs, FGDs and observations (n = 1) [41]; and photovoice (n = 2) [46,47]. Three studies employed mixed methods [43,48,52]. However, some included studies referred to other chronic conditions such as stroke, gout, asthma, depression, sickle cell anemia, epilepsy, and chronic obstructive pulmonary disease. These conditions were only mentioned in the context of co-occurrence with our priority diseases but were not analyzed in our synthesis. Their presence in the reviewed studies highlights the broader multimorbidity landscape but did not influence the scope of our analysis.

### Synthesis of findings

We carried out a meta synthesis review on self-management of multimorbidity among patients living with a combination of HIV, diabetes, hypertension or chronic kidney disease. We synthesized findings under four main inductive themes namely: i) medical management; ii) diet management; iii) emotional management; and iv) physical management.

## Medical management

**Biomedical treatment is prioritized, but access is limited.** The papers reviewed reported high awareness of the need for biomedical treatment among patients and carers. Commonly practiced medical management strategies include active search for medical information [15,41,50]; maintenance of medical appointments and drug adherence [23,33,34,37,41–43,45,51–53]; self-monitoring [15,41,42,46,48,51]; and avoidance of trigger factors, for example wearing tight shoes for diabetic patients [33,38,42]. Biomedical management of chronic conditions was considered central towards optimal disease control, even more so when implemented alongside dietary and spiritual management [15,23,31–34,36,37,41,47,49,54]. Patients living with diabetes fear serious consequences of hypoglycemia or hyperglycemia and claim increased treatment compliance than the other diseases [41]. The immediate and potentially fatal effects of blood sugar fluctuations create significant psychological stress, which in turn reinforces adherence to treatment as a coping mechanism to prevent severe complications [33,41,54]. PLWMM are more likely to implement biomedical management to prevent secondary complications [36,41,54].

> *"I am always afraid of low blood sugar because I have experienced it on several occasions. I go about with cube sugar in case of sudden low blood sugar. I also take cola drink whenever I have low blood sugar which I do with caution to avoid elevated blood sugar level'. (FGD 4, 48-year-old Civil servant)* [48].

> *"BP* (hypertension) *is dangerous because if people have not gone to the hospital, they may suffer a stroke or lose the use of some limbs or even die. Sometimes we hear of people just dying without getting sick,"* (male, 58, 13 years on ARTs, never misses ARTs- also antihypertensive).

Patients living with singular disease experience more treatment satisfaction compared to PLWMM, reflections on treatment effectiveness relate to initiation of treatment and are directly linked to reduced distress and improved functionality [15,36,39,52].

> *"Before I started taking my medication my heart could beat faster. In the case of diabetes, before I had knowledge of it I couldn't cultivate* (farm) *because I felt very weak and I couldn't figure it out that it was due to diabetes, but after I started taking medication now I can cultivate for long time [...] unlike before,"* Female, 55 diabetes and hypertension [15].

> *"...the time when I did not know of my status...I was very sick not expecting to be alive up to date like the way I am.... I was not able to eat, to talk to anyone, or even to sit like this, no! I could only sleep and maybe do my toilet right where I was. But just after I started taking my medicine...I see that I am getting much better...,"* IDI Female, HIV [52].

> *"A lot has changed in my life because at first, I did not have good health like this. My life was tough because I was suffering a lot and lost weight but when I got diagnosed and initiated on treatment I have seen great improvement in my general health and I have even gained more weight than before I started treatment,"* Man, diabetic [39].

Treatment adherence was also described in reference to experiencing symptoms. Sometimes participants mentioned poor adherence to treatment especially when symptoms of disease are absent.

> *"I know it's important, but sometimes I stop taking the medicine when I feel fine, then I realize that was a mistake." Male, hypertensive* [50].

Post initiation of treatment, most PLWMM observe and monitor how their bodies function living with disease [23,31,34,41,42,48,52]. Patients living with HIV comorbidity are more likely to pay attention to the other disease than the HIV as many perceive their HIV to be well-managed [23,31,32,34,41,42,52]. Again, symptom recognition related to

diabetes or hypertension is a more complicated process than in HIV. This may also demonstrate that patient experiences of comorbidity may be more subjective or prolonged for diabetes, hypertension and chronic kidney disease than for HIV itself.

**Rationing drug supply as a strategy to manage access to drugs**

Medication stock-outs and high costs are reported in all studies [15,23,31–34,36,37,40–43,48,49,51,52]. Regular drug stockouts likely indicate that shortage of medication for chronic conditions in SSA is a regional challenge. Patients and their carers negotiate such challenges in multiple ways. Most studies indicate that NCD clinics are not integrated, requiring PLWMM to make multiple trips for each disease. For patients with HIV comorbidity, most papers report that patients access ART care nearby and visit a hospital further away for the other disease like diabetes or hypertension. Although ART care is generally decentralized to primary health systems, this review highlights that lack of integrated and coordinated care at tertiary facilities compounds burden of treatment through increased time and financial demands for PLWMM [37,39,52].

> *"I always get my ARVs from Ntabiseng clinic in Bara, and my diabetes pills from the diabetes clinic," female living with HIV and diabetes* [23].

> *"Transportation is difficult because I am not working. I attend four clinics [...]. Today I am here, then next month I'll attend two other clinics." (Person living with multimorbidity)* [23].

While challenges persist, one study reported enhanced access to information, health education and medication availability improved diabetes self-management amongst patients in Malawi [39]. This study aimed to explore self-management of diabetes at a facility implementing the World Health Organization's Package of Essential NCD Interventions for primary health care in low-resource settings (WHO PEN-Plus) [39].

We found that patients prioritize obtaining treatment despite financial concerns. Patients perceive the health outcomes of missing diabetes and hypertension treatment to outweigh the financial related costs due to transport, thus informing the decision making process [43,52,55]. Out-of-pocket expenditure thus indicates a strong commitment to self-management among patients and their carers. However, the desire to maintain adherence and lack of access to treatment presents a dilemma to PLWMM, one that defines levels of adherence. Some patients modify treatment schedule and ration drugs titrated against symptoms of disease, a practice observed particularly among hypertensive patients [36,40,52], who may also purchase medications from informal sources like street vendors due to lower costs [52]. Some patients, however, cannot afford medications representing a significant strain on financial well-being [36,42]. This leads to delayed or absent treatment, potentially leading to disease complications [43,50], while others may turn to alternative treatment pathways, such as traditional care [23,42,43,51,52] or apply dietary management as an alternative, rather than complimentary treatment [51]. Apart from limited access to biomedical treatment, the need to access care from multiple providers is also described by patients or carers who expressed dissatisfaction with biomedical treatment; perceived that treatment was too slow; or experienced side effects [23,42,49,52]. This requirement to access multiple providers may be associated with decreased use of biomedical care.

> *"OK, we've tried the Western type of medicine for a long time...but it's not getting better as we would want it; we want see the other aspect the traditional and the religious one, see whether it will help him to be fine, even if he will be able to stand on the other foot." (Guardian to a diabetic single amputee)* [40]....*"*

In some instances, patients may favor spiritual or traditional management over biomedical care but these indicate change in self-management after experiencing disease deterioration [23,36,41,42,51,52]. In such cases, biomedical treatment is

sometimes sought after experiencing an acute event [41,42], some adopting exclusive medical treatment plans to minimize future disease complications [23,41]. However, because of the belief that diabetes is caused by spiritual means, one study insisted that patients relegated biomedical health care to the background in favor of spiritual care throughout most of the patients' journey [44].

Multiple medical appointments lead to increased financial costs for patients [23,31,32], including failure to make medical appointments for fear of confrontation by health workers for seeking care outside the allocated day [23,31,32]. Sometimes traditional vertical models of care delivery for PLWMM result in contradictory information regarding handling of conditions and decreased confidence in health care worker capacity to address complex cases [15,23,31].

> "The problem is that one doctor will tell you to do this and another asks you to do a different thing. I see different doctors for each disease. Last week, the Rheumatologist told me that my bones are getting closer to each other, they have inserted metals in my right foot. When I attended the diabetes clinic [earlier], the doctor asked me to exercise because I was adding more weight, but I can't exercise because of the surgery they did on my leg. My ARVs have amplified my appetite. I eat a lot and I am worried about my weight too. Last time I asked the endocrinologist what else I could do to deal with my appetite, my weight gain and inability to exercise, but he just looked at me without saying a word." IDI, Female, 64, HIV, HTN, DM and arthritis [23].

### Mistrust presents barriers to implementation of biomedical care

Poor communication around the implementation of medical management for disease may create mistrust among patients and their carers. Mis- or incomplete information can impact on medication adherence amongst patients or carers, presenting a significant barrier to biomedical care [33,51]. Treatment non-adherence emerges as partial, with deliberate on-and-off uptake, or complete non-uptake, with alternatives such as diet or spiritual management implemented as stand-alone forms of care [51].

> "Why do they give me free medicines for this diabetes? They know that it will never be cured. A real medicine is never free." Diabetes patient [33]

> "I have purchased medicines that were fake... it is known that some make fake drugs so they can make lots of quick cash." (Diabetic individual) [51].

> "When they prescribe me pharmaceutical drugs, I refuse, because I have heard that once you are used to the medicine, you have to take them every day," (Diabetic individual) [51].

One study reports community delivery of diabetes drugs in South Africa improved access to drugs [31]. This adds on to evidence that availability of drugs closer to home improves treatment adherence [31,39].

### Stigma associated with chronic disease

Qualitatively, stigma was noted and reported differently across HIV and NCDs [15,44]. HIV stigma was often linked to deep-rooted moral judgment, blame, and social rejection, reflecting the intense and dehumanising nature of HIV stigma. In some cases, disclosure of conditions was secondary to maintenance of social relationships especially among individuals living with HIV. Thus the value assigned to one's social identity may negatively affect treatment adherence [15,23,44,49,52] and potentially increase the risk of disease complications [31,52] and social exclusion [52].

> "...when I was very sick, my [relatives] said 'you AIDS person, is this not AIDS? why are you not dying?'" IDI Male HIV 005 [31].

In contrast, diabetes became a source of stigma not inherently, but due to community misconceptions linking it with HIV [15,44]. Diabetes-related stigma appeared more context-specific and commonly affected younger individuals, who were sometimes assumed to be HIV-positive due to the belief that diabetes is a condition of older age.

*"People even regard diabetes as HIV/AIDS, and therefore diabetics hide to avoid stigmatisation. The people say diabetes is for the elderly … and when young people get diabetes, they think they have HIV,"* Female 43 caregiver, urban [44].

*"Others were speaking in a mockery way saying this person suffers from diabetes and saying some words. I was worried with those words and would ask myself how come [...] my friends are talking about me and mocking me,"* Male KR<50 (Hypertension, Diabetes) [15].

### Dietary self-management practices

Studies highlight the importance of dietary management among PLWMM but that implementation is impacted by poverty and high cost of food [15,33,37,39,40,43,45,54]; limited knowledge on portion size estimates [37,51]; and complexity of conditions and patients' contrary cultural values [34,37,49,51]. Studies highlight that preparation of separate meals for patients with chronic disease [34,51,52] is challenging, citing poverty and increased workload related to food preparation as the main reasons for not following dietary requirements [34,37,41,48,49,52].

Patients and carers understanding of diet management are varied [44–47]. Lay descriptions of recommended dietary practices include reducing the consumption of carbohydrates or sugar. However, these practices indicate faulty understanding of pertinent issues to mean only sweet foods contain sugar or starch is only present in flour. It also highlights limited comprehension of the need to balance amounts and time of consumption. Some patients and their carers suggest budgetary prioritization to optimize dietary management strategies [47].

*"Sometimes these days I do not eat breakfast because I think after dinner I'm satisfied until lunch."* (diabetic patient) [51] or not eating *'starch'* but eating *'rice'* (diabetic patient) [40].

*"Oil that settles off when placed down can give you the illness […],"* (42-year-old diabetic patient) [41].

*"If I want to eat the rice, my wife put the rice on a fire and put water on it, she will let the rice boil. Then she takes the starch and waste it. She put clean water on the rice again and wash it. Then she wastes it and then she let the rice steam. It shouldn't be soft, it should be, you know, just normal rice. Then the rice does not contain no more starch,"* man, diabetic, 62, retired college graduate [47].

The inaccurate interpretations regarding what is starch or sugar, or the physical presentation of cooking oil stress that knowledge on what, how, when or how much negatively affects dietary implementation for disease [41,44,51]. The challenge of comprehending dietary recommendations can create confusion when advice suggests moderating rather than completely eliminating certain foods [23,36].

*"From what they said, it looks like everything related to eating habits. We eat early before going to bed, so I don't eat at night. I don't take cassava and corn dough, but I eat akple (a staple maize meal) and a lot of fruits and vegetables instead of consuming sugary foods,"* Female 60, diabetic [44].

*"When I go to the private clinic the doctors advise me to stop taking meat, salt, sweet beverages, and not to use cooking fat/oil, but without telling me the reasons. When I come to the public hospital the nurse explains that I should take food in moderation, so I don't know who to believe,"* Female 58, overweight, diabetic [37].

Diet management also relates to social factors, kinship, cultural beliefs and systems indicating an intricate relationship between cultural beliefs and what people eat [34,37,46,47,49,51]. We note that cultural identity, family influence, or community norms may significantly impact dietary practices, sometimes leading to a preference for traditional foods that do not align with medical dietary recommendations. Two studies indicate patient' implementation of contrary practices indicative of the value placed in cultural identity [33,49].

> *"Being African, I eat rice for lunch because it is part of my culture and Senegalese men like to eat fatty and oily "thiebou dien" every day* (Diabetic patient) [49].

Desperation or defeatist attitude emerge among diabetic patients who feel hopeless [33,49,51] citing diabetes: *"diabetes is a slow poison"* [37,51]. Such an attitude may normalize certain behaviors contrary to recommended practices for PLWMM. Alternatively, it may indicate a lack of assertiveness or demotivation among patients, who may feel overwhelmed. This perspective can also weaken patients' resolve to adhere to their treatment plans if they perceive their efforts as futile against the challenges posed by their condition.

> *"There are several people that I have met who do not care [about diabetes]. During social events such as wedding, prayers, or funeral, you will see them overeat... they tell me... you only die once."* (Man, diabetic) [51].

> *" You cannot tell your wife that these days do not prepare this. Because you are the only one with diabetes.... You must be flexible... the issue of dietary restrictions is problematic."* (Diabetic man) [51].

**Psychological stress among PLWMM**

Patients and carers within the included studies reported that anxiety and depression are more pronounced among PLWMM than with single chronic disease [15,23,31,33,35,36,43,52]. Patients with both diabetes and chronic kidney disease had particularly high levels of stress [15,23]. Studies reported that a single diagnosis of hypertension or HIV were associated with lower stress levels [52]. Reported stressors included pain; deterioration of disease; onset of new illness and symptoms. These were exacerbated by poverty and limited access to biomedical treatment. For effective emotional management of chronic disease, patients must be able to risk assess; manage time; communicate; and think analytically. These processes facilitate problem solving; allocation of scarce resources; prioritization of conditions; and coordination of self-care [35–37,52]. In most cases, the dominant sources of information include family members (including those who live or had provided care to family member with a similar disease), other patients and health workers [15,23,41,56]. The studies demonstrate that PLWMM avoid stressors that may complicate their conditions [15,31,36,37,41].

> *"I must also keep my temper in check as I can't afford to lose it otherwise, I risk escalating my blood sugar level and blood pressure".* (Woman with hypertension and diabetes) [31].

Depression is cited by PLWMM as they negotiate the burden of conditions [23,43] or disrupted social relationships [15,34,52]. Participant emphasized that living with multimorbidity was overwhelming and that family support plays a central part in managing the psychological burden of the conditions.

> *"In fact, if you don't have a strong family support, you would be humiliated because everything about diabetes and hypertension involve money…if you don't have anyone in the family to support and always be close to you, you will deteriorate. Because at a point, if you don't get support financially and physically, you will die from stress and depression,"* Man with comorbid T2D and hypertension [43].

### Lay understanding and implementation of physical activity

PLWMM and their carers recognise the benefit of regular physical activity towards functional performance [15,32,44,50]. Some studies highlight the link between physical management and well-being [15,34].

*"I bought a testing device which I use to check myself every day. When I was employed, it was helping me because I was walking a long distance to minibus depot which was part of exercise and with the nature of my work, I was working under the sun and I was sweating, so it was also part of my physical exercise and when I test my sugar the following morning it was at a good level such as 90,"* Male <50 hypertension and diabetic [15].

For women living with multimorbidity, physical exercise are embedded within daily living activities such as domestic chores like cleaning, washing and walking to work [37,42,48] rather than public exercise [23,42,51]. This integration of physical exercise into daily living activities offered practical, accessible, and sustainable approaches to improving health outcomes for PLWMM [37,44,52].

Barriers to physical management for PLWMM are pain, age, lack of clarity on nature of exercise and competing priorities [15,31,33,34,41,42,45,50,52]. Anxiety, fear of exertion and motivation affect implementation of physical exercise [23,48]. Fear stems from concerns about worsening their condition, experiencing pain, or facing physical limitations, thereby creating a cycle of avoidance and inactivity that perpetuates disease complications [34].

*"… with me I don't walk long distances at all because of my chest, I get a tight chest when I walk a distance and I start coughing,"* Patient with hypertension [32].

*'I do try to exercise sometimes, but you know I get tired … and I will do it for a month only, then I just get tired. I am not lazy, but ….'* Female, hypertension [32].

In one study, physical management is applied as an alternative in the absence of diet management.

*'To deal with these challenges, Participant Y has opted to pay for a gym, which he finds to be affordable at R340 (US$22) a month, compared with buying healthy foods which he says "are expensive."'* Man, 67, HTN and DM [23].

### Social capital for managing multimorbidity

Social support is reported to facilitate self-management in all included studies [23,33,34,41–43,50–52,55]. Where disclosure of condition(s) is accepted, the patient or carer leverages on social capital and gets support for financial, physical or emotional management [15,35–37,41]. Conversely, lack of social support means patients learn to manage other people's expectations [35,49,54], and sometimes, self-isolate where community expectations contradict recommended medical care [15,48]. Failure to manage external expectations sometimes lead to feelings of worthlessness among patients [15,33]. Challenges to emotional support include competing priorities among individuals who can provide support, family and social commitments and PLWMM themselves being carers to others living with chronic conditions [19,28,30]. The absence of emotional support negatively affects relationships [15,31,52]. Peer support is common among patients with HIV, whether as singular or comorbid conditions [23,31,34,36,37,41,48,52].

### Discussion

These results offer a comprehensive understanding of the experiences and coping strategies related to self-managing encompassing self-management of the HIV, diabetes, hypertension and chronic kidney disease, and multimorbidity in any combination of these conditions among patients and their carers in SSA. This is important because effective self-management strategies

are key to controlling chronic conditions, preventing complications, and improving quality and longevity of life. Further work is required to explore how patients and carers can be empowered to manage the four chronic conditions or multimorbidity. We recommend that health literacy programmes are tailored to context to ensure that recommendations are sustainable and deliverable within the healthcare system and that advice is tailored toward social and cultural norms and practical realities.

We found high intention to use biomedical treatment among patients with multimorbidity or individual chronic diseases in SSA [15,23,31–34,36,37,40–43,48–52]. However, multiple factors limit implementation of effective self-management strategies. In SSA, patients frequently must access multiple providers and incur out-of-pocket expenses to obtain medications, increasing treatment burden and financial strain [16,43,50,57,58]. Financial constraints require that PLWMM often need to ration drugs or compromise dietary choices. This highlights an unresolved compliance-affordability paradox: while many patients, especially those with diabetes, express heightened fear of complications and strong motivation to adhere to treatment, this intention is undermined by limited access to medication, diagnostic services, and appropriate dietary options and financial resources, which drive treatment rationing or symptom-based medication titration. Clearly, working toward universal healthcare coverage (UHC) would help to address some of these issues. This includes leveraging existing HIV care infrastructure such as integrating NCD drug supply chains, counseling and laboratory services and community health programs which would offer a cost-effective and immediately actionable step to improve service delivery and reduce financial burden on patients [59,60]. Whilst health ministries develop and implement UHC, however, pragmatic sustainable guidance should be developed for patients to help them make optimal prioritisation decisions based on their individual circumstances. Work to improve health literacy and advocacy in patients and their carers is important to drive self-efficacy and self-management of multimorbid disease with examples in Malawi [61], Ethiopia [62], Uganda [63] and South Africa [16,20]. Building social support facilities may also help PLWMM and their carers navigate perceived barriers to self-management [64]. In Kenya, for example, PLWMM who attended support groups demonstrated improved decision making and treatment compliance [65].

Our meta-synthesis shows that PLWMM and their carers apply a biosocial lens to self-management and implement strategies between conditions according to their perceived understanding of disease etiology and severity. A systematic review on determinants of HIV treatment adherence in SSA reports that adherence is linked to patient perceptions towards improved health outcomes [66]. A key priority, therefore, for PLWMM in SSA would be to link management strategies with measures of disease control. Creating these feedback loops would help to reinforce self-management decisions and priority setting for both patients and their carers. For example, temporary drug withdrawal was commonly reported among patients with hypertension. This may reflect fear of long-term treatment [65,67] and/or communication challenges between healthcare workers and patients/carers [67]. Measures to increase patient autonomy and ownership of their hypertension together with a disease control feedback loop (e.g., blood pressure control measurements) could promote improved self-management decisions. Strategies to improve physical activity in PLWMM need to be tailored toward individuals so that they are meaningful but also relevant and sustainable. Limited knowledge and low self-efficacy are currently reported as barriers to physical exercise for PLWMM within the current literature [25,32,57,68].

## Strength and limitations

To our knowledge, this study is the first qualitative meta-synthesis review to synthesize lived experiences regarding self-management of multimorbidity among patients living with a combination of diabetes, hypertension, chronic kidney disease or HIV. Our authorship group represents a multidisciplinary team including both clinical and qualitative expertise to facilitate rigorous meta-synthesis approaches and clinical relevance. The strength of our findings are limited by the relative paucity of relevant literature in this area. Exclusion of studies not published in English is a limitation and may reduce the generalisability of our findings across different linguistic and cultural contexts. We excluded one study published in Portuguese [69]. However, we acknowledge that language limitation may introduce potential bias and generalizability of findings to non-English contexts. Despite evidence that most a majority of articles are published in English (96% of all PubMed

indexed articles published in English in 2015, 89% between 2000–2020) [69], the exclusion of non-English articles might still exclude significant studies. We did not include quantitative data from studies that utilised mixed methods within our qualitative synthesis. As a result, the review may overlook important numerical insights that could contribute to a more comprehensive understanding of the topic. Future reviews should consider methodological triangulation to integrate both qualitative and quantitative findings for a more holistic analysis. We also aimed to capture self-management of multimorbidity across SSA, but a geographical imbalance exists, with more studies from Southern and parts of Eastern Africa and limited representation from Central and West Africa. This reflects the availability of published research rather than exclusion of certain regions, highlighting the need for more studies in underrepresented areas. Additionally, this may limit the generalization of findings to Francophone or West African contexts.

To ensure that interventions are not only theoretically sound but also practical and effective in real-world settings, it is important to assess programs using scientifically proven frameworks. Future programs can draw lessons from the Integrated Chronic Care Clinic (IC3 model) currently piloted at a facility in rural Southern Malawi [70] which has demonstrated the feasibility of leveraging existing HIV infrastructure for NCD care while operating with minimal additional financial or human resources. We also propose evaluating such programs in the context of multimorbidity and patient-centered care in SSA using the RE-AIM framework (Reach Effectiveness Adoption Implementation and Maintenance) [71,72]. By using a mixed methods approach, this would support a holistic assessment of interventions by examining patient engagement, impact on health outcomes and behaviors, provider uptake, implementation fidelity, and long-term sustainability within health systems and communities [72]. Particular attention should be paid to how integration models account for and respond to condition-specific stigma, ensuring that care delivery is not only clinically appropriate but also socially responsive and inclusive [73]. Such an evaluation is critical to ensure that programs effectively address the complex needs of PLWMM and support sustainable self-management across diverse contexts.

## Conclusion

We found that PLWMM and their carers were motivated to self-manage their conditions but face multiple barriers including both financial and health systems constraints. Health systems strengthening is essential to develop more horizontal and holistic models of care delivery to reduce the need for multiple healthcare interactions and promote joined up care. Contextually sensitive measures to improve health literacy and autonomy should be prioritised as a component of this strengthening to help patients and their carers develop the tools required to manage their long-term health conditions. The development and support of community groups may bolster this process where members can support delivery of appropriate lay information and provide support to patients, reducing pressure on health systems.

## Supporting information

**S1 Table. Search strategy.**
(DOCX)

**S2 Table. Summary of studies included in the review.**
(DOCX)

**S3 Table. CASP checklist.**
(DOCX)

**S4 Table. List of all studies screened.**
(DOCX)

**S5 Table. List of all full studies screened.**
(DOCX)

**S1 Fig. PRISMA flow chart diagram.**
(TIF)

**S1 Checklist. PRISMA checklist.**
(DOCX)

## Acknowledgments

We would like to acknowledge Ms. Alison Derbyshire, the Academic Liaison & Training Specialist, Liverpool School of Tropical Medicine (LSTM) for her assistance in the search strategy. We would also like to acknowledge the authors of the original manuscripts for their dedication to this topic and for elevating the voices of marginalized people in sub-Saharan Africa.

## Author contributions

**Conceptualization:** Sangwani Nkhana Salimu, Melissa Taylor, Nicola Desmond, Deborah Nyirenda, Ben Morton.

**Data curation:** Sangwani Nkhana Salimu, Melissa Taylor, Stephen A. Spencer, Nicola Desmond, Deborah Nyirenda, Ben Morton.

**Formal analysis:** Sangwani Nkhana Salimu, Melissa Taylor, Stephen A. Spencer, Deborah Nyirenda, Ben Morton.

**Methodology:** Sangwani Nkhana Salimu, Melissa Taylor, Stephen A. Spencer, Nicola Desmond, Ben Morton.

**Supervision:** Nicola Desmond, Deborah Nyirenda, Ben Morton.

**Writing – original draft:** Sangwani Nkhana Salimu.

**Writing – review & editing:** Melissa Taylor, Stephen A. Spencer, Nicola Desmond, Deborah Nyirenda, Ben Morton.

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
