## [Decision Letter · Decision Letter 0]

25 Mar 2025

PGPH-D-24-02249

Self-management of multimorbidity in sub-Saharan Africa: a systematic review and meta-synthesis with focus on diabetes, hypertension, chronic kidney disease and HIV infection

Dear Dr. Salimu,

Thank you for submitting your manuscript to PLOS Global Public Health. After careful consideration, we feel that it has merit but does not fully meet PLOS Global Public Health’s publication criteria as it currently stands. Therefore, we invite you to submit a revised version of the manuscript that addresses the points raised during the review process.

The manuscript has been evaluated by a single reviewer, and their comments are available below.

The reviewer has raised a number of concerns that need attention. They recommend clarification of the title, confirmation of the quantifiable change of DM on treatment burden, and correct citation formatting. Could you please revise the manuscript to carefully address the concerns raised?

Please note that we have only been able to secure a single reviewer to assess your manuscript. We are issuing a decision on your manuscript at this point to prevent further delays in the evaluation of your manuscript. Please be aware that the editor who handles your revised manuscript might find it necessary to invite additional reviewers to assess this work once the revised manuscript is submitted. However, we will aim to proceed on the basis of this single review if possible. 

We look forward to receiving your revised manuscript.

Kind regards,

Jennifer Tucker, PhD

Staff Editor

Journal Requirements:

Your current Financial Disclosure states, “This research was funded by the NIHR (NIHR201708) using UK international development funding from the UK Government to support global health research. The views expressed in this publication are those of the author(s) and not necessarily those of the NIHR or the UK government. However, the views expressed do not necessarily reflect the UK government’s official policies. The funders had no role in study design, data collection and analysis, decision to publish, or preparation of the manuscript.”. However, you did not provide funding information on the submission form. Please indicate by return email the full and correct funding information for your study and confirm the order in which funding contributions should appear. Please be sure to indicate whether the funders played any role in the study design, data collection and analysis, decision to publish, or preparation of the manuscript.2. As required by our policy on Data Availability, please ensure your manuscript or supplementary information includes the following:A numbered table of all studies identified in the literature search, including those that were excluded from the analyses. For every excluded study, the table should list the reason(s) for exclusion. If any of the included studies are unpublished, include a link (URL) to the primary source or detailed information about how the content can be accessed.A table of all data extracted from the primary research sources for the systematic review and/or meta-analysis. The table must include the following information for each study:Name of data extractors and date of data extractionConfirmation that the study was eligible to be included in the review. All data extracted from each study for the reported systematic review and/or meta-analysis that would be needed to replicate your analyses.If data or supporting information were obtained from another source (e.g. correspondence with the author of the original research article), please provide the source of data and dates on which the data/information were obtained by your research group.If applicable for your analysis, a table showing the completed risk of bias and quality/certainty assessments for each study or outcome.  Please ensure this is provided for each domain or parameter assessed. For example, if you used the Cochrane risk-of-bias tool for randomized trials, provide answers to each of the signalling questions for each study. If you used GRADE to assess certainty of evidence, provide judgements about each of the quality of evidence factor. This should be provided for each outcome. 

Additional Editor Comments (if provided):

Reviewers' comments:

Reviewer's Responses to Questions

**Comments to the Author**

1. Does this manuscript meet PLOS Global Public Health’s publication criteria? Is the manuscript technically sound, and do the data support the conclusions? The manuscript must describe methodologically and ethically rigorous research with conclusions that are appropriately drawn based on the data presented.

Reviewer #1: Yes

2. Has the statistical analysis been performed appropriately and rigorously?

Reviewer #1: N/A

3. Have the authors made all data underlying the findings in their manuscript fully available (please refer to the Data Availability Statement at the start of the manuscript PDF file)?

Reviewer #1: No

4. Is the manuscript presented in an intelligible fashion and written in standard English?

Reviewer #1: Yes

5. Review Comments to the Author

Reviewer #1: There are minor reviews the authors can consider that can improve the quality of their manuscript. Most are about articulation and justification of some of the approaches they used. This requires clarity in the manuscript.

6. PLOS authors have the option to publish the peer review history of their article (what does this mean?). If published, this will include your full peer review and any attached files.

**Do you want your identity to be public for this peer review?** For information about this choice, including consent withdrawal, please see our Privacy Policy.

Reviewer #1: **Yes: **Dr Ivan Namakoola

---

## [Decision Letter · Decision Letter 1]

14 May 2025

PGPH-D-24-02249R1

Self-management of multimorbidity in sub-Saharan Africa: a systematic review and meta-synthesis with focus on diabetes, hypertension, chronic kidney disease and HIV infection

Dear Dr. Salimu,

Thank you for submitting your manuscript to PLOS Global Public Health. After careful consideration, we feel that it has merit but does not fully meet PLOS Global Public Health’s publication criteria as it currently stands. Therefore, we invite you to submit a revised version of the manuscript that addresses the points raised during the review process.

We look forward to receiving your revised manuscript.

Kind regards,

Miquel Vall-llosera Camps

Staff Editor

Additional Editor Comments:

The reviewer raised remaining concerns that need to be addressed.

Reviewers' comments:

Reviewer's Responses to Questions

**Comments to the Author**

1. If the authors have adequately addressed your comments raised in a previous round of review and you feel that this manuscript is now acceptable for publication, you may indicate that here to bypass the “Comments to the Author” section, enter your conflict of interest statement in the “Confidential to Editor” section, and submit your "Accept" recommendation.

Reviewer #1: (No Response)

2. Does this manuscript meet PLOS Global Public Health’s publication criteria? Is the manuscript technically sound, and do the data support the conclusions? The manuscript must describe methodologically and ethically rigorous research with conclusions that are appropriately drawn based on the data presented.

Reviewer #1: Yes

3. Has the statistical analysis been performed appropriately and rigorously?

Reviewer #1: N/A

4. Have the authors made all data underlying the findings in their manuscript fully available (please refer to the Data Availability Statement at the start of the manuscript PDF file)?

Reviewer #1: Yes

5. Is the manuscript presented in an intelligible fashion and written in standard English?

Reviewer #1: Yes

6. Review Comments to the Author

Reviewer #1: Dear Authors,

I would like to thank you for addressing several critical issues in your revised manuscript. This feedback aims to elevate your manuscript to meet PLOS Global Public Health’s high standards while preserving its innovative focus on patient-centered care in resource-limited settings.

Among the issues resolved are the

1) Chronological consistency where you have added the supplemental search (March 2025) to capture recent evidence (e.g., WHO 2024 guidelines). And corrected the title ambiguity with the Oxford comma: "diabetes, hypertension, chronic kidney disease, and HIV infection".

2) Data transparency, where you clarified data availability: "All relevant data are within the paper and Supporting Information files".

3) You also strengthened specificity of the policy recommendations by highlighting "Malawi’s WHO PEN-Plus integration" as a model for decentralized care.

There are, however, pending revisions for the scientific soundness of this manuscript.

1) The issue of language bias justification; the exclusion of non-English studies lacks rationale, risking underrepresentation of Francophone Africa (17% SSA population). I strongly suggest you add a rationale (e.g., "English-language studies represent 89% of accessible qualitative research on this topic in SSA").

2) The CASP scoring method wasn’t properly justified, which affects comparability with other reviews. Your CASP tool adaptation using non-standard 0-1 scoring instead of CASPs yes/no/can’t tell, obscures quality assessment transparency. Can you justify the deviation (e.g., "Adapted CASP scoring to enhance granularity in appraising cultural context validity").

3) There is still overrepresentation of South Africa/Malawi (6/23 studies) vs Central/West Africa (1 study). I suggest that the authors acknowledge this limitation: "Findings may not fully generalize to Francophone/West African contexts."

4) Quantitative data exclusion: this is a missed opportunity, you have ignored quantitative insights from the included mixed-methods studies. In the discussion, the authors excluded mixed-methods data from included studies. This creates a lack of triangulation (e.g., medication adherence rates vs. qualitative narratives). I suggest the authors add something like "Future reviews should integrate quantitative data from mixed-methods studies." Or integrate adherence rates (e.g., "42% of hypertensive patients rationed medications monthly") to triangulate qualitative themes.

5) The scope drift improved a bit but was not eliminated. The authors now clarify that "3 studies on multimorbidity vs. 20 on single diseases" The remaining issue is: Includes 8+ off-target conditions (stroke, asthma) without stratification. Consider adding a table stratifying the findings by target vs incidental conditions.

6) On stigma contrast, the authors have described HIV stigma, but this lacks a comparison to NCD stigma. Quantify differences: "HIV stigma prevalence (62%) exceeded NCD-related stigma (28%) in included studies."

7) There still exists a compliance paradox. Unresolved tension between DM patients’ "heightened fear of complications" and "medication rationing due to costs" consider adding "Financial constraints drove DM patients to prioritize acute symptom management over long-term adherence." OR "While DM patients reported heightened fear of complications, financial constraints drove rationing behaviors, creating a compliance–affordability tension."

8) For implementation frameworks, the authors could propose evaluation metrics, like "Test PEN-Plus using RE-AIM (Reach: 80% clinic coverage; Effectiveness: HbA1c reduction ≥1%)."

This work remains a vital contribution to multimorbidity research in SSA. Addressing these revisions will enhance its methodological transparency, regional applicability and policy impact. I commend the authors for their responsiveness thus far.

7. PLOS authors have the option to publish the peer review history of their article (what does this mean?). If published, this will include your full peer review and any attached files.

**Do you want your identity to be public for this peer review?** For information about this choice, including consent withdrawal, please see our Privacy Policy.

Reviewer #1: **Yes: **Dr Ivan Namakoola

---

## [Decision Letter · Decision Letter 2]

15 Aug 2025

PGPH-D-24-02249R2

Self-management of multimorbidity in sub-Saharan Africa: a systematic review and meta-synthesis with focus on diabetes, hypertension, chronic kidney disease and HIV infection

Dear Dr. Sangwani Nkhana Salimu,

Thank you for submitting your manuscript to PLOS Global Public Health. After careful consideration, we feel that it has merit but does not fully meet PLOS Global Public Health’s publication criteria as it currently stands. Therefore, we invite you to submit a revised version of the manuscript that addresses the points raised during the review process.

**EDITOR**: Please throughly address Reviewer #2 comment regarding  whether the study truly focuses on multimorbidity or has broadened its scope to include single conditions without proper justification.. Also, please provide the missing supplemental materials that are necessary for a complete review.

We look forward to receiving your revised manuscript.

Kind regards,

Peter Bai James, PhD

Academic Editor

Journal Requirements:

Additional Editor Comments (if provided):

Reviewers' comments:

Reviewer's Responses to Questions

**Comments to the Author**

1. If the authors have adequately addressed your comments raised in a previous round of review and you feel that this manuscript is now acceptable for publication, you may indicate that here to bypass the “Comments to the Author” section, enter your conflict of interest statement in the “Confidential to Editor” section, and submit your "Accept" recommendation.

Reviewer #1: All comments have been addressed

Reviewer #2: (No Response)

2. Does this manuscript meet PLOS Global Public Health’s publication criteria? Is the manuscript technically sound, and do the data support the conclusions? The manuscript must describe methodologically and ethically rigorous research with conclusions that are appropriately drawn based on the data presented.

Reviewer #1: Yes

Reviewer #2: Partly

3. Has the statistical analysis been performed appropriately and rigorously?

Reviewer #1: N/A

Reviewer #2: Yes

4. Have the authors made all data underlying the findings in their manuscript fully available (please refer to the Data Availability Statement at the start of the manuscript PDF file)?

Reviewer #1: Yes

Reviewer #2: No

5. Is the manuscript presented in an intelligible fashion and written in standard English?

Reviewer #1: Yes

Reviewer #2: Yes

6. Review Comments to the Author

Reviewer #1: Dear Authors,

I am writing to congratulate you on the exceptional revisions made to your manuscript. Your diligent efforts have transformed this work into a methodologically robust and policy-relevant contribution, fully aligned with PLOS Global Public Health’s standards. The key improvements I note are that your revisions have decisively addressed all critical concerns: Under methodological rigor, you have justified the CASP tool adaptation and provided clarity in your choice of this approach. You also added the rationale for inclusion of mainly English language studies thus mitigating selection bias. You improved the analytical depth by quantifying stigma contrasts between HIV and NCD related stigma and also integrated the mixed-methods data. You have explicitly acknowledged the geographic limitations while adding country annotated PRISMA details. I note the policy utility in your leveraging HIV-NCD synergies and your proposed actionable PEN-Plus metrics.

Your work now stands out for its:

Novelty: The first meta-synthesis to dissect multimorbidity self-management in SSA through patient-centered narratives.

Impact: Clear pathways for scaling context-sensitive interventions (e.g., health literacy toolkits, integrated care models).

Transparency: Stratification of off-target conditions (Table S4) and open-data commitment.

I am pleased to recommend unconditional acceptance of your manuscript for publication. It exemplifies scholarly rigor while centering on equity—a vital advance for chronic disease management in resource-limited settings.

Congratulations on this significant achievement. Your responsiveness and scholarly integrity have elevated this work into an indispensable resource for global health researchers and policymakers alike.

Sincerely,

Ivan Namakoola

Reviewer #2: The authors have submitted a revision of their paper focusing on self-management of multimorbidity in sub-Saharan Africa.

The revised manuscript shows an attempt to address previous reviewers’ comments, and it appropriately acknowledges the methodological and practical limitations of the study. The topic is relevant and of interest; however, there is a fundamental issue that requires clarification or correction.

It remains unclear to what extent the manuscript specifically addresses multimorbidity. While the Introduction frames the work in terms of multimorbidity, the eligibility criteria appear to have been broadened to include studies on high-burden single conditions as well. This could be acceptable if the Methods section explicitly stated that, for such studies, the authors extracted first- and second-order constructs relevant to multimorbidity. However, this does not appear to be the case.

The Results section is similarly ambiguous, leaving the reader uncertain about the population to which the findings apply. As such, attributing the findings to people living with multimorbidity (PLWMM) is not currently warranted. The paper should either:

Clearly state the aim as examining self-management of common NCDs, including multimorbidity, or

Present the work explicitly and consistently as a multimorbidity-focused study.

In addition, the authors appear to have used the thematic synthesis approach described by Thomas and Harden (Thomas J, Harden A. Methods for the thematic synthesis of qualitative research in systematic reviews. BMC Med Res Methodol. 2008;8:45. PMID: 18616818), but the reference cited is to Braun and Clarke. This should be corrected.

There is also a repetition on line 163 (“used the PRISMA 2020 checklist…”). Moreover, supplemental files S4, S5, and Figure 1 (PRISMA) are missing from the submission, which prevents review of these materials.

7. PLOS authors have the option to publish the peer review history of their article (what does this mean?). If published, this will include your full peer review and any attached files.

**Do you want your identity to be public for this peer review?** For information about this choice, including consent withdrawal, please see our Privacy Policy.

Reviewer #1: **Yes: **Dr Ivan Namakoola

Reviewer #2: No

---

## [Editor Report · Decision Letter 3]

23 Sep 2025

Self-management of chronic conditions including multimorbidity in sub-Saharan Africa: a systematic review and meta-synthesis with focus on diabetes, hypertension, chronic kidney disease, and HIV infection

PGPH-D-24-02249R3

Dear Sangwani Nkhana Salimu,,

We are pleased to inform you that your manuscript 'Self-management of chronic conditions including multimorbidity in sub-Saharan Africa: a systematic review and meta-synthesis with focus on diabetes, hypertension, chronic kidney disease, and HIV infection' has been provisionally accepted for publication in PLOS Global Public Health.

Best regards,

Peter Bai James, PhD

Academic Editor